# Coral larval aquaculture: Species-specific survival and microbial dynamics in flow-through systems

**Blake D. Ramsby**[1,2]*, **Ramona Brunner**[3], **Jonathan Barton**[1], **Sophie Ferguson**[1], **Clare Grimm**[1], **Yilmaz Can Hiçyilmaz**[1,2,3], **David G. Bourne**[1,2,3], **Andrew P. Negri**[1], **Andrea Severati**[1], **Yui Sato**[1], **Muhammad Azmi Abdul Wahab**[1]

**1** Australian Institute of Marine Science, Townsville MC, Townsville, Queensland, Australia, **2** AIMS@ JCU, Townsville, Queensland, Australia **3** College of Science and Engineering, James Cook University, Townsville, Queensland, Australia

* b.ramsby@aims.gov.au

⬅ OPEN ACCESS

## Abstract

To enable burgeoning reef restoration efforts, the production of coral larvae in captive aquaculture needs to maximize survival and yields. While current coral larval rearing methods are generally successful, the approaches vary, and larval yields can be limited due to gradual declines in larval quality or sudden crashes. Here, we reared *Acropora* sp. nov. aff. *kenti* and *Acropora spathulata* larvae for seven days under different stocking densities (0.3, 1.0, or 2.0 larvae mL$^{-1}$), water volume turnover rates (0.2 or 0.6 vol. hr$^{-1}$), and with or without UV sterilization and surface agitation, to evaluate their effects on larval survival. Overall, culture treatments had minimal impact on the high larval survival of *A.* sp. nov. aff. *kenti* after seven days in culture. However, *A. spathulata* declined to less than half of the stocking density regardless of culture treatment. Other larval performance metrics including visual appearance and size were not different between the culture treatments. Microbial community composition of the culture water, assessed through 16S rRNA gene sequencing, showed varied community structure between turnover treatments for *A.* sp. nov. aff. *kenti* but not for *A. spathulata*, suggesting species-specific or dynamic responses to water exchange rates. Larval culture water quality parameters (including dissolved and particulate C and N) were consistent across treatments, although reduced water turnover briefly lowered nitrate and nitrite concentrations, coinciding with elevated particulate carbon. This correlated with an increase in potential denitrifying bacteria within the *Gammaproteobacteria*, suggesting longer residence times of culture water can increase microbial activity. These results demonstrate that large-volume flow-through systems are a robust method for achieving high larval yields for some species and underscore the importance of understanding species-specific survival dynamics to optimize coral aquaculture for large-scale reef restoration.

**Data availability statement:** The microbial 16S sequences used in community analysis are available via NCBI BioProject PRJNA1306316 (http://www.ncbi.nlm.nih.gov/bioproject/1306316). All other data are available via the AIMS Data Repository (https://apps.aims.gov.au/metadata/view/88ccf845-3c40-4e9c-b4b9-f7f7bf0f3e86).

**Funding:** This research was supported by the Reef Restoration and Adaptation Program (RRAP), which aims to develop effective interventions to help the Reef resist, adapt, and recover from the impacts of climatechange, and which is funded by a partnership between the Australian Government's Reef Trust and the Great Barrier Reef Foundation.

**Competing interests:** The authors have declared that no competing interests exist.

## Introduction

Frequent and repeated disturbances driven by anthropogenic pressures are compromising the structure and function of coral reefs [1,2] and, in response, prompting efforts to develop and implement effective strategies to maintain the resilience of coral reef ecosystems [3]. Scaling up restoration methods, such as harnessing mass coral spawning to produce corals in captive aquaculture, is required to supply corals to support reef restoration [4]. However, coral spawning in captive systems faces several challenges, including optimizing fertilization, maintaining genetic diversity, sustaining large larval populations, and improving the survival of young corals [5]. While techniques to isolate gravid colonies, collect gametes, and fertilize eggs are well established [6,7], more work is needed to identify culture conditions that maximize larval yields [8]. Previous research has focused more on post-settlement survival than on mortality during the larval phase [9,10], partly because broadcast spawning corals can produce vast numbers of larvae [11]. However, as restoration efforts scale up, increasing the efficiency of larval rearing is essential, with improved early-stage survival required to meet the rising demand for larvae and reduce production costs. There is a need for systematic evaluation of larval densities and tank conditions to enhance yield, while minimizing water use and labor [8].

Rearing density is a key factor influencing aquaculture efficiency [12,13]. While coral larvae are routinely cultured at densities between 0.2–1.5 larvae mL$^{-1}$, few studies have directly compared density treatments. Low densities (<0.3 larvae mL$^{-1}$) may improve survival in large tanks (e.g., 45% in 500 L cultures; [8]), but higher densities may be feasible due to the low metabolic demands of larvae [14]. Coral larvae have been shown to survive at 4 larvae mL$^{-1}$ in small closed containers (15–200 mL; [12]), and natural slicks can contain up to 10 larvae mL$^{-1}$, albeit with survival below 20% [15]. To maximize production, direct comparisons of larval survival and performance across culture densities are needed.

Water quality is another critical determinant of larval culture success [5]. Water exchange maintains oxygen levels, removes waste, and can control temperature, as high temperatures can lead to abnormal larval development [16]. High larval survival (70%) is achievable after three days without water exchange at 4 larvae mL$^{-1}$ [12], but the physicochemical tolerances of coral larvae remain poorly understood. Excess sperm, nitrate accumulation, and associated water quality issues have been implicated in larval mortality [8,17], therefore identifying the minimum water turnover required to maintain larval health could streamline aquaculture operations, especially in resource-constrained settings.

In addition to water chemistry, microbial dynamics may strongly influence larval survival. Declining cultures often appear cloudy, oily, or foamy [6], suggesting bacterial proliferation, linked to decaying larvae or unfertilized eggs [18,19]. Once visible signs of microbial fouling appear, the collapse of culture, defined as rapid mortality of larvae, is typically irreversible [6]. Although antibiotics can suppress pathogenic bacteria such as *Vibrio* spp. [19–21], their use is often impractical at scale and

undesirable for flow-through systems. Whether sterilization or high-water turnover can limit bacterial blooms and improve survival remains largely untested.

Despite the ability of current captive coral breeding programs to produce hundreds of thousands of larvae [5,22], further improvements in efficiency are needed to scale production for meaningful reef restoration outcomes. In this study, we evaluated the survival of two species of *Acropora* larvae reared in flow-through cultures under a range of culture conditions, including variable stocking densities, water turnover rates, UV sterilizations, and surface agitations. Larval survival, size and appearance alongside water quality and microbial community diversity and abundance were tracked over a 7-day culture period. Additionally, we examined larval settlement success to determine whether rearing conditions affected larval competency. Our findings demonstrate that flow-through culture systems support high larval survival across a broad range of conditions, with successful rearing up to at least 1 larvae mL$^{-1}$.

## Methods

### Coral collection and larval production

Two separate experiments were performed to test the effect of culture treatments on *Acropora* sp. nov. aff. *kenti*, and on *Acropora spathulata* (Brook, 1891). The former species is closely related to *A. kenti* (Brook, 1892), but further taxonomic work is needed to resolve this cryptic group of species [23]. Here, we refer to *A.* sp. nov. aff. *kenti* as *A. kenti* for brevity.

Gravid *A. kenti* colonies were collected in November 2022 from the Palm Island Group, QLD, Australia (18°45'54.8"S, 146°31'36.1"E), while gravid *A. spathulata* colonies were collected in December 2022 from Davies Reef QLD, Australia (18°49'6.9"S, 147°38'49.2"E). Colonies of both species were collected using a hammer and chisel on SCUBA (1–9 m depth; GBRMPA Permit G21/45348.1) and transported to the Australian Institute of Marine Science National Sea Simulator (Townsville, QLD, Australia), where they were held in outdoor aquaria (27.4 °C in November and 28.3 °C in December, max. ~200 µmol quanta m$^{-2}$ s$^{-1}$). Holding systems were 2800 L recirculating systems (280 × 100 × 44 cm) that received 3 turnovers of filtered seawater (FSW) per day. After 6–9 days in holding, colonies with emergent bundles of egg and sperm were isolated. Bundles of sperm and egg were collected from the water surface above each colony within 1 h of release by skimming using plastic cups. Bundles were then divided amongst three 80 L flow-through tanks where bundle separation and fertilization was performed following Severati et al. [7]. The final concentration was ~26 zygotes mL$^{-1}$ with >80% of zygotes cleaved after 1.5 h for both species. All zygotes were then distributed amongst the 18 culture treatment tanks described below. There was no evidence that unfertilized eggs affected culture quality as no signs of dead biomass or steep declines in larval density were observed after 12–24 hrs.

### Larval culture treatments

Stocking of the culture treatments tanks differed slightly for the *A. kenti* and *A. spathulata* experiments due to the number of larvae produced during each spawning. For *A. kenti*, 6.1 million total zygotes from 11 colonies were fertilized in three tanks on the same night, pooled, and then distributed among 18 culture tanks at densities of 0.3 or 1.0 larvae mL$^{-1}$ (Table 1).

In contrast, 4.2 and 3.0 million *A. spathulata* zygotes were obtained over two consecutive spawning nights and fertilized in one tank on each night. Fifteen tanks were stocked (either 0.3 or 1.0 larvae mL$^{-1}$) using larvae from 12 colonies on the first night, while 3 tanks were stocked (2.0 larvae mL$^{-1}$) the following night using larvae from 13 colonies due to an insufficient number of larvae from the first night. Four colonies repeatedly spawned on both nights. For *A. spathulata,* larvae from each fertilization tank were allocated equally across culture tanks without pooling.

Each 500 L treatment tank received flow-through FSW (1 µm) from four angled inlets at the water surface, providing clockwise water rotation (Fig 1). Water exited the tank via a central drainpipe (30 cm × 5 cm each at half of the depth of

**Table 1. Culture treatments for *Acropora kenti* and *Acropora spathulata*, including stocking density, turnover, UV sterilization (Steril.), and surface agitation.**

| Species | Target density | Turnover | Steril. | Surface agitation | Survival | Size and App. | Microb. sampling (n) |
|---|---|---|---|---|---|---|---|
| | (larvae mL⁻¹) | (vol. hr⁻¹) | | | (n) | (n) | |
| *A. kenti* | 0.3 | 0.2 | – | – | 3 | 3 | 0 |
| | 0.3 | 0.2 | UV | – | 3 | 3 | 3 |
| | 1 | 0.2 | – | – | 2 | 3 | 0 |
| | 1 | 0.2 | UV | – | 2 | 3 | 3 |
| | 1 | 0.6 | UV | – | 3 | 3 | 3 |
| | 1 | 0.6 | UV | Air | 1 | 3 | 0 |
| *A. spathulata* | 0.3 | 0.2 | – | – | 3 | 3 | 0 |
| | 0.3 | 0.2 | UV | – | 3 | 3 | 3 |
| | 1 | 0.2 | – | – | 3 | 3 | 0 |
| | 1 | 0.2 | UV | – | 3 | 3 | 3 |
| | 1 | 0.6 | UV | – | 3 | 3 | 3 |
| | 2 | 0.6 | UV | – | 0 | 0 | 0 |

A hyphen is used to indicate where sterilization or surface agitation was not used. For each treatment, the number of tanks (n) analyzed is listed for each larval survival, larval size and appearance, and microbial sampling. Several tanks were not included for analysis of larval survival due to losses of larvae from clogged filters.

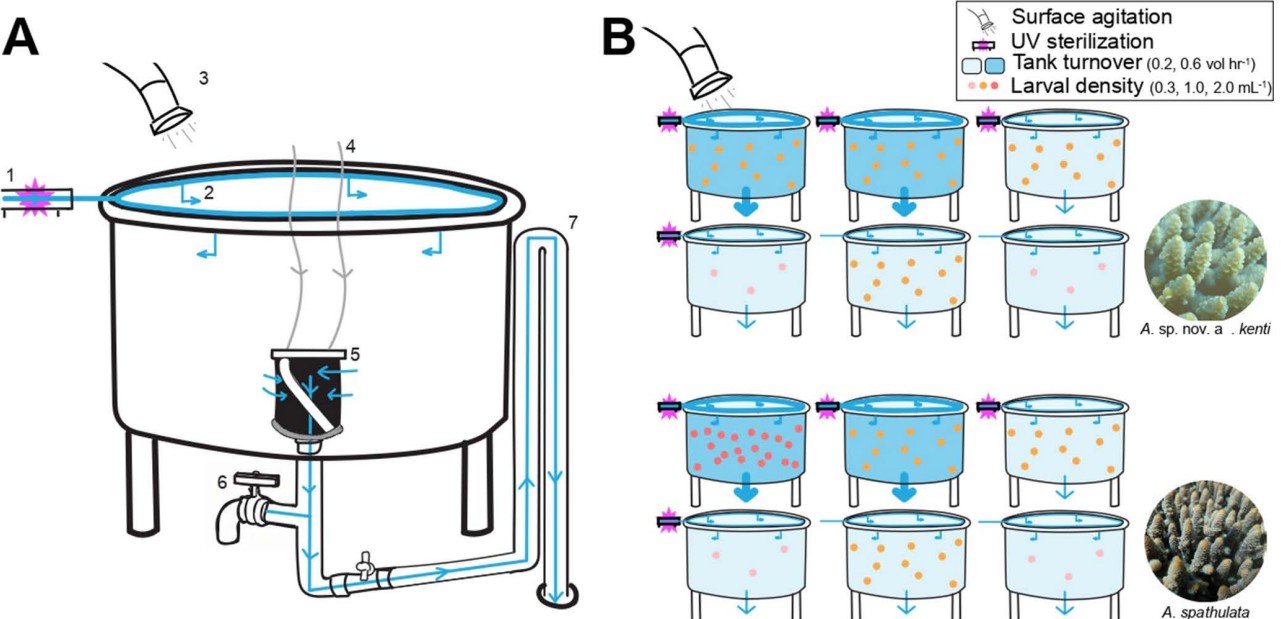

**Fig 1. Culture tank design and experimental overview.** A) A diagram of a 500 L culture tank highlighting the culture treatments: (1. UV sterilization, 3. air-driven surface agitation) and tank features (2. Angled inlets generating circular rotation, 4. Aeration delivered to the bottom of the tank, 5. Cylindrical mesh filter on outlet, 6. Water sample outlet, 7. Standpipe to set operational height). Water flow is indicated using blue arrows, air flow is indicated using grey arrows. B) A visual representation of the six combinations of culture treatments used for each species (see Table 1). Unique symbols are used to denote air-driven surface agitation and UV sterilization as depicted in A). The tank color indicates the tank turnover (i.e., flowrate) with darker blue indicated 0.6 vol. hr⁻¹. The pink, orange, and dark orange circles represent a larval density of either 0.3, 1.0, or 2.0 larvae mL⁻¹ stocking density, respectively.

the tank) with four 220 μm mesh panels. The drainpipe was connected to an external standpipe that maintained the water depth in the tank. At the bottom of the drainpipe, a foam ring bubbled air to prevent larvae contacting the outlet mesh. Water temperature was controlled using a supervisory control and data acquisition system (Siemens SCADA PCS7, Germany) that mixed 22 and 36 °C seawater to reach target temperatures. Water flow to each tank was controlled using ball valves. *A. kenti* cultures were kept at 27.4°C while *A. spathulata* cultures were kept at 28.3°C (±<0.1°C SE) to match reef temperatures at the times of collection (S1 Fig.). Culture tanks received only low irradiance from the fluorescent lights in the facility (~5 μmol m$^{-2}$ s$^{-1}$).

A total of 18 treatment tanks were assigned to six experimental treatments, each representing a combination of larval density, tank turnover, UV sterilization of incoming water, and surface agitation (n = 3 tanks per treatment). Five of the six treatments were consistent across both species, incorporating the same combinations of larval density, turnover rate, and UV sterilization (Table 1). Two larval densities (0.3 or 1.0 larvae mL$^{-1}$) were tested to determine whether stocking density affected larval survival or water quality by increasing metabolic waste as previously suggested [8,19]. Two water flow rates (100 or 300 L h$^{-1}$, equivalent to 0.2 or 0.6 vol. hr$^{-1}$) were used to assess whether higher turnover improved culture performance while UV sterilization of incoming seawater (SMART UV sterilizer E120S 120W; Pentair, Australia) was also assessed. The sixth treatment differed between species with *A. kenti* cultures subjected to surface agitation via four air blowers installed above each tank to reduce embryo accumulation along tank walls, a known source of mortality (Abdul Wahab pers. obs.). Gentle aeration (20 L min$^{-1}$) was applied during the first 24 hours to avoid damaging early-stage zygotes [24], followed by increased airflow (100 L min$^{-1}$) for the remainder of the culture period. For *A. spathulata,* the sixth treatment tested a higher larval density (2.0 larvae mL$^{-1}$) to assess the potential for increasing larval yield under intensive culture conditions.

Larval cultures were maintained under these treatments for up to 7 days post-fertilization. Cultures were inspected daily for signs of larval deterioration, including surface foam or larval mortality. Culture maintenance was performed as required, including gentle stirring to break up clumps of larvae and removal of surface biofilms using a paper towel (as per [25]). All manual culture maintenance and handling was consistent across all tanks and treatments.

The measurement intervals for larval characteristics, water quality, and microbial communities are presented in S1 Table.

## Larval characteristics

Larval density, size, and gross morphology were recorded daily from each tank. Larval density was measured daily from a 1 L water sample from each tank that was collected at the water surface after gentle homogenization, filtered through a 212 μm mesh to concentrate the larvae in 70 mL. Larvae were subsequently counted in 3 mL subsamples of the resuspension using a Bogorov chamber (n = 5 counts per subsample). Larval survival was calculated as a proportion of the nominal stocking density for each treatment (0.3, 1.0, or 2.0 larvae mL$^{-1}$).

To measure larval size and assess morphological changes, a sample of approximately 18 larvae per tank were photographed daily in a thin layer of water in a petri dish to keep larvae in a flat plane with a 14MP camera (LC3CMOS; TOUPCAM, P. R. China) mounted on a dissecting microscope (MS5; Leica Microsystems GmbH, Germany). Larval size was measured from these images using Ilastik software [26]. Briefly, larvae were identified via trained pixel-based segmentation, and their area was calculated in mm$^2$. Additionally, an Ilastik object classifier was trained to distinguish larval shapes as round, elongated, or abnormal (S2 Fig.). Following image segmentation, a size filter (0.01 < size < 0.15 mm$^2$) was applied to exclude abnormal or non-larval objects. A median of 18 larvae (range: 1–64 larvae) was analyzed per tank per day and totalled 30–90 larvae per treatment per day. The proportion of morphologically 'normal' larvae (i.e., round or elongated) was calculated relative to the total number of larvae observed.

Larval competency for settlement was assessed on day 5 of the culture, within the competency window for multiple *Acropora* species [27]. For each tank, larvae (n = 10) were transferred into 12 wells of sterile 6-well culture plates, each

containing 10 mL FSW and a small (~25 mm$^2$) fragment of the crustose coralline algae (CCA) *Porolithon* cf. *onkodes*, a known settlement inducer for *Acropora* spp. [28]. An additional 12 wells without CCA served as negative settlement controls. After 24 h, the number of larvae that had settled and metamorphosed was recorded to quantify settlement success.

## Water quality

Dissolved oxygen (DO), pH, and temperature were measured daily at midday using a HQ30D multi-parameter meter (Hach, USA). The probes were calibrated at the beginning of the 7-day measurement period according to the manufacturer's instructions.

Water samples were collected from each culture tank on Day −1 (i.e., prior to larval stocking) and Days 1, 2, 4, and 6 post-stocking with zygotes. Samples were taken from the outflow valve, downstream of a mesh filter, to prevent larval loss during collection. All samples were collected in sterile containers. For dissolved nutrient analysis, 10 mL of water was filtered through 0.45 µm Minisart Cellulose Acetate filters (Sartorius, Germany). Samples for dissolved organic carbon (DOC) and total dissolved nitrogen (TDN) were acidified with 100 µL 37% hydrochloric acid and analysed using a TOC-L analyser (Shimadzu, Japan). Duplicate samples for dissolved inorganic nitrogen (DIN: $NH_4^+$, $NO_2^-$, $NO_3^-$), free reactive phosphate ($PO_4^{3-}$), and reactive silica (Si) were measured using a AA500 segmented flow analyser (SEAL Analytics, Germany). To measure particulate organic carbon (PC) and particulate nitrogen (PN), 250 mL of water was filtered onto pre-combusted glass fibre filters (Whatman, United Kingdom). Filters were acidified with 37% hydrochloric acid and analysed on a TOC-V analyser (Shimadzu, Japan) equipped with a total nitrogen unit and solid sample module. Samples for PC, PN, DIN, and $PO_4^{3-}$ were stored at −20°C until analysis. Samples for DOC and TDN were stored at 4°C prior to analysis.

## Microbial community of culture water

Microbial sampling focused on three of the six culture treatments for each coral species to examine effects of stocking density and tank turnover on microbial dynamics. A sample of the seawater supplying the culture tanks was included for comparison. Samples were taken prior to stocking (Day −1) and on Days 1, 2, 4 and 6 of culture. For each sampling point (n = 3 per treatment and n = 1 seawater source), four replicates of 50 mL seawater were collected and filtered onto sterile 0.22 µm Millex-GP Syringe Filters (Merck Millipore, Germany) using a vacuum manifold. To each filter, 144 µL of 1.25 × lysis buffer was added for lysozyme digestion and sample preservation following the DNeasy protocol (Qiagen, Germany). Filters were stored at -20 °C until DNA extraction.

DNA was extracted using a modified DNeasy Blood & Tissue Kit (Qiagen, Germany) protocol to initiate cell digestion directly within the syringe filter. The frozen filters were thawed and incubated with lysozyme for 60 min at 37°C. Lysate was flushed from the filters with 1 × lysis buffer and collected in new tubes. Filters were then incubated with Proteinase K and AL buffer for 60 min at 56°C, after which lysate was flushed into the sample tube using AL buffer. DNA was extracted from the lysate following the kit protocol and stored at 4°C.

To quantify total bacteria and *Vibrio* spp. abundance, multiplex droplet digital polymerase chain reaction (ddPCR) was used to target 16S ribosomal RNA (16S rRNA) from all four filters per sample. Each reaction consisted of 12.5 µL ddPCR EvaGreen Supermix (Bio-Rad, USA), 250 nM forward and reverse broad-range bacterial/archaeal primers (1406F/1525R; [29]), 400 nM forward and reverse *Vibrio* spp.-targeting primers (Vib1-f 567F/Vib2-r 680R; [30]), and 1 µL DNA template. Droplets were generated using a AutoDG and Droplet Generation Oil for EvaGreen (Bio-Rad, USA). The thermal profile was run at 95°C for 10 min, 40 cycles of 95°C for 30 sec, 56°C for 1 min, 98°C for 10 min, followed by an infinite hold at 4°C. Droplet fluorescence was measured using a QX200 droplet reader (Bio-Rad, USA) and analysed in QuantaSoft™ Analysis Pro software. Fluorescence thresholds were identified on the ddPCR profiles to distinguish amplification of *Vibrio* spp. versus bacterial/archaeal 16S rRNA sequences using profiles of *Vibrio* positive controls in each run. To compare bacterial abundance in cultures to local reefs, water samples were collected adjacent to coral colonies (3–5 m depth) at the

Palm Island Group, Davies Reef, and Magnetic Island (19°9'19.1'' S, 146°51'56.9'' E) in November 2022, and processed as described above. Each filter or field sample was measured twice and averaged to calculate the bacterial and *Vibrio* abundance.

To compare bacterial community composition across culture treatments and with the seawater source, 16S rRNA amplicon sequencing was performed on extracted total DNA from two of the four filters per sample (Ramaciotti Centre for Genomics, UNSW, Australia). The V3-4 region of 16S rRNA was amplified with primers 341f and 805r [31] and sequenced using Illumina MiSeq, resulting in 2 × 300 bp paired-end reads that were merged. Demultiplexed and adapter-trimmed sequences were processed in QIIME2 [32].

Sequences from two filters per sample were pooled after preliminary inspections showed high similarity (totaling 8604–345182 sequences per sample; mean 152910). Sequence quality was assessed with FASTQC v 0.12.1 and MultiQC v 1.13a [33,34] and further quality-filtered, checked for chimeras, paired reads joined, and trimmed in dada2 using default parameters (reduced to 3574–174041 sequences per sample; mean 79628). Amplicon sequence variants (ASV) were identified by comparing sequences against a SILVA database (version 138.1) after low quality sequences (>5 ambiguous bases, homopolymer length >8) and short sequences had been removed (Archaea >900 bp, Bacteria >1200 bp, Eukarya >1400 bp) from the reference set. Sequences were dereplicated to 99% identity [35,36]. Only sequences matching bacterial Phyla references were analyzed after excluding 15,782 sequences belonging to Chloroplasts, Mitochondria, Eukaryota, Archaea and Unassigned sequences. The data set was then reduced to 17,903 sequences per sample to account for differences in sampling depth. Three samples with less than 17,903 sequences were excluded from analysis.

## Statistical analysis

All statistical analyses and graphical results were performed in R version 4.3.0 [37]. Each response variable was analyzed separately for each species using generalized linear mixed models with culture treatment (the combinations of larval densities, turnovers, sterilizations, and surface agitations listed in Table 1), time, and the treatment*time interaction as predictors [38]. All analyses included a random intercept for each culture tank except PERMANOVA of microbial communities. Different error distributions were used for each response variable including gaussian (larval survival, size, water quality variables, and Faith's phylogenetic diversity), binomial (proportional appearance and settlement), and negative binomial distributions (bacterial and *Vibrio* abundance). Model residuals were inspected visually for normality and homogeneity of variance (gaussian models) and compared observed residuals to simulated residuals to (binomial and negative binomial models; [39]).

Post-hoc analyses were designed to identify treatments that affected the coral larvae or the culture environment and on which days any effects occurred. Pairwise comparisons were performed among all treatments using the Tukey method [40], however many culture treatments differed in multiple parameters (e.g., stocking density and turnover). To ease interpretation and take a conservative statistical approach, only comparisons differing in a single culture parameter are discussed (e.g., stocking density only). For significant treatment*time interactions, post hoc comparisons were used to identify significantly different levels of parameters on each day (e.g., 0.3 vs 1.0 larvae mL$^{-1}$ with all other factors being equal; S2 Table). Significant effects that occurred on multiple days or in both coral species were considered strong evidence of a treatment effect. When the treatment*time interaction was insignificant, any significant main effects of culture treatment or time were compared for each day in the same way. For main effects of culture treatment, we focused only on pairwise comparisons that differed in a single factor. For main effects of time, the pairwise comparisons over time were done sequentially (e.g., Day 1 vs Day 2, Day 2 vs Day 3, and so on) to determine when changes occurred.

ASV diversity was calculated using Faith's phylogenetic diversity index [41], representing the sum of phylogenetic branch lengths of unique sequences within a sample, using QIIME2 software. Differences in bacterial community compositional were assessed using Bray-Curtis distances in the *vegan* package in R [42]. Differences in microbial community composition between turnover treatments and over time were analyzed using permutational analysis of variance and

9,999 permutations [42]. Community distances between turnover and sampling days were visualized using a principal coordinates analysis.

## Results

At the highest density and turnover rate tested (2 larvae mL$^{-1}$ and 0.6 vol. hr$^{-1}$), mesh filters clogged, causing overflows and losses of *A. spathulata* larvae. No data could be collected from this treatment, and it was excluded from analyses. In addition, four *A. kenti* tanks were excluded from survival analysis due to equipment failure that caused larval losses through mesh filter leakage on the outflow which was detected during water sampling. Final sample sizes for each treatment are presented in Table 1.

### Larval characteristics

The accuracy of larval stocking varied between species. For *A. kenti*, measured larval densities on day 2 were higher than the nominal stocking values: 0.4 (±0.02 SE) and 1.2 (±0.05 SE) larvae mL$^{-1}$ for the nominal 0.3 and 1.0 larvae mL$^{-1}$ treatments, respectively. In contrast, for *A. spathulata*, day 2 densities were lower than the nominal: 0.2 and 0.8 larvae mL$^{-1}$ for the nominal 0.3 and 1.0 larvae mL$^{-1}$ treatments, respectively.

   *A. kenti* larvae maintained high survival (~100% of the targeted stocking density) over the 7-day culture period, whereas *A. spathulata* survival declined to less than half (~40%) by Day 7 across all treatments (Fig 2A). For both species, survival was not affected by culture treatments (Table 2). For *A. kenti*, no two consecutive timepoints showed significantly different survival, indicating relatively stable culture performance. In contrast, *A. spathulata* exhibited significant declines from Day 3 to day 4 (85 to −71%) and again from Day 5 to Day 6 (63–49%), regardless of treatment (S3 Table). By Day 7, *A. spathulata* survival averaged 40±3%, compared to 110±4% for *A. kenti* (Fig 2A).

   On average, *A. spathulata* larvae were 25% larger than *A. kenti* larvae (0.076±0.002 versus 0.060±0.001 mm$^2$; Fig 2B). For both species, larval size varied over time depending on culture treatment (treatment*day; Table 2). In *A. kenti*, no individual treatment caused significant differences in larval size on any day. For *A. spathulata*, larval size was significantly influenced by stocking density on Day 4, with larvae reared at 0.3 larvae mL$^{-1}$ being larger than those at 1.0 larvae mL$^{-1}$ (S4 Table). No effects of turnover rate, UV sterilization, or surface agitation were detected on larval size for either species.

   Most larvae appeared morphologically normal, with 86±1% of *A. kenti* and 75±3% of *A. spathulata* displaying a round or elongated shape (Fig 2C). For *A. kenti*, the proportion of normal larvae varied with culture treatments and timepoint independently but showed no significant treatment*time interaction (Table 2). Pairwise comparisons revealed no significant effects of any treatment on larval appearance (S5 Table). However, larval size in *A. kenti* increased significantly from Day 3 to Day 4 (S4 Table). For *A. spathulata,* larval appearance varied over time according to culture treatment (Table 2), with significantly more larvae appearing normal at a turnover rate of 0.2 versus 0.6 vol. hr$^{-1}$ on Days 5 and 6 (S4 Table). Stocking density, UV sterilization, and surface agitation had no detectable effects on larval appearance.

   Larval settlement differed markedly between species. *A. kenti* larvae had higher settlement success (81±2%) than *A. spathulata* (45±4%; S3 Fig). Culture treatments significantly affected larval settlement outcomes for both species (Table 2), though the significant effects differed. In *A. kenti*, settlement was 18% higher in cultures without UV sterilization compared to those with UV sterilization. In contrast, UV sterilization did not affect settlement in *A. spathulata* but *A. spathulata* reared in low-turnover tanks exhibited significantly higher settlement (by 21%) compared to those in high turnover conditions (S5 Table). Stocking density and surface agitation had no significant effect on settlement success. No settlement was observed for the negative controls without CCA present.

### Water quality

Although differences in water quality were observed among culture treatments, two consistent patterns emerged in both *A. kenti* and *A. spathulata* cultures. Firstly, stocking density was negatively associated with NO$_3^-$ and turnover was positively associated with both NO$_2^-$ and NO$_3^-$ concentrations, although the differences were small (~0.3 µmol; Fig 3A, Fig 3B). While

 

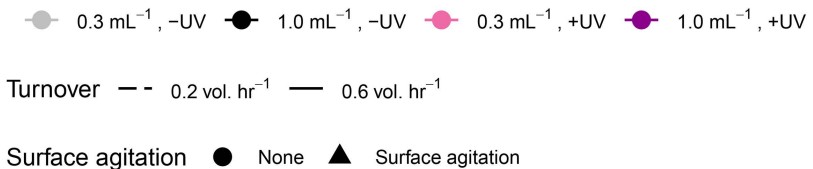

**Fig 2. Temporal patterns in larval cultures.** A) Larval survival (proportion of target stocking density) for *A. kenti* and *A. spathulata* in different culture treatments. B) Larval size (area mm²). C) Proportion of larvae with normal morphologies (i.e., round or elongated). Columns distinguish coral species. Larval stocking densities are represented using different colors, tank turnover treatments have dashed or solid lines, and the surface agitation treatment is denoted with triangles. Points represent means and SE.

**Table 2. Statistical results comparing culture responses across different culture treatments over time.**

| | Response | Day in culture | | | Treatment | | | Treatment * Day | | |
|---|---|---|---|---|---|---|---|---|---|---|
| | | df | Test stat. | p | df | Test stat. | p | df | Test stat. | p |
| *A. kenti* | Survival | 5 | F = 6.5 | **<0.01** | 5 | F = 0.6 | 0.68 | 25 | F = 0.8 | 0.68 |
| | Size | 5 | F = 127.5 | **<0.01** | 5 | F = 2.1 | 0.14 | 25 | F = 1.6 | **0.04** |
| | Appearance | 5 | $X^2$ = 16.5 | **0.01** | 5 | $X^2$ = 11.1 | **0.05** | 25 | $X^2$ = 33.9 | 0.11 |
| | Settlement | – | – | – | 5 | $X^2$ = 13.0 | **0.02** | – | – | – |
| | Bact. abundance | 4 | $X^2$ = 156.8 | **<0.01** | 2 | $X^2$ = 14.0 | **<0.01** | 8 | $X^2$ = 80.1 | **<0.01** |
| | *Vibrio* abundance | 4 | $X^2$ = 25.0 | **<0.01** | 2 | $X^2$ = 3.7 | 0.16 | 8 | $X^2$ = 25.6 | **<0.01** |
| | Faith's phyl. div. | 4 | F = 23.4 | **<0.01** | 1 | F = 46.1 | **<0.01** | 4 | F = 0.4 | 0.80 |
| *A. spathulata* | Survival | 5 | F = 47.6 | **<0.01** | 4 | F = 1.5 | 0.28 | 20 | F = 1.0 | 0.43 |
| | Size | 6 | F = 9.8 | **<0.01** | 4 | F = 3.6 | **0.02** | 23 | F = 2.6 | **<0.01** |
| | Appearance | 5 | $X^2$ = 10 | 0.07 | 4 | $X^2$ = 8.9 | 0.06 | 20 | $X^2$ = 49.2 | **<0.01** |
| | Settlement | – | – | – | 5 | $X^{2=}$24.5 | **<0.01** | – | – | – |
| | Bact. abundance | 4 | $X^2$ = 9.6 | **<0.01** | 2 | $X^2$ = 0.3 | 0.84 | 8 | $X^2$ = 35.1 | **<0.01** |
| | *Vibrio* abundance | 4 | $X^2$ = 13.5 | **0.01** | 2 | $X^2$ = 0.7 | 0.69 | 8 | $X^2$ = 15.4 | **0.05** |
| | Faith's phyl. div. | 4 | F = 30.8 | **<0.01** | 1 | F = 5.0 | 0.09 | 4 | F = 0.9 | 0.51 |

Survival, density, size, and Faith's phylogenetic diversity were analyzed using generalized linear mixed models and the degrees of freedom (df), test statistic (F), and p value are reported. Appearance, settlement, bacterial abundance, and *Vibrio* abundance were analyzed using generalized linear mixed models (test statistic = $X^2$). P-values ≤0.05 are indicated in bold.

larval density may have contributed to the $NO_3^-$ depletion, reduced turnover appeared to have greater influence. For instance, cultures at 1 larvae mL$^{-1}$ with 0.2 vol. hr$^{-1}$ exhibited transient $NO_3^-$ depletion of >1.0 µmol, whereas cultures at the same density but with 0.6 vol. hr$^{-1}$ maintained $NO_3^-$ levels comparable to the source water (Fig 3B). In both species, significant reductions in $NO_3^-$ were limited to two Days (S6 Table, S7 Table). Other dissolved nutrients (TDN, $NH_4^+$, Si, TDP, and DOC) occasionally differed between treatments, but the patterns were inconsistent between species (S2 Fig). Overall, there was little evidence that increasing stocking density to 1 larvae mL$^{-1}$ or reducing flow-through turnover to 0.2 vol. hr$^{-1}$ adversely affected water quality.

Secondly, $NO_3^-$ depletion coincided with changes in particulate N and particulate C (Fig 3C, Fig 3D). On the same Days that $NO_3^-$ concentrations declined, PN and PC levels increased in high density, low turnover cultures of both species (1 larvae mL$^{-1}$, 0.2 vol. hr$^{-1}$), exceeding concentrations in the source water (Fig 3). Smaller increases in PN and PC were observed in high-turnover cultures (0.6 vol. hr$^{-1}$), indicating that particulate accumulation was influenced by turnover rate. UV sterilization and surface agitation had no consistent or strong effects on water quality (S6 Table, S7 Table). For *A. kenti*, UV sterilization reduced $NO_2^-$ and $NO_3^-$ (Day 2 only), increased DOC (Day −1 and Day 1), and elevated TDP (Day 1; S4 Fig., S6 Table). Surface agitation reduced $NO_2^-$ and $NO_3^-$ concentrations on Days 1 and 2 (S6 Table). In *A. spathulata*, UV sterilization reduced $NH_4^+$, $NO_2^-$, $NO_3^-$, TDP, and Si on one day during the culture period (S4 Fig., S7 Table). Interestingly, UV sterilization was consistently associated with 0.2 °C cooler water temperatures compared to the non-UV treatments across both species (S7 Table).

For most nutrient parameters, differences between culture treatments varied significantly over time (Treatment*Day, Table 3). However, the temporal patterns differed between the two *Acropora* species. In *A. kenti,* almost all significant treatment-related differences in nutrient concentrations occurred within the first 2 days, whereas in *A. spathulata,* nearly all differences emerged between Days 4 and 6 (Fig 3, S6 Table, S7 Table). In both cases, significant effects on water quality were transient, occurring on fewer than half of the Days (S6 Table, S7 Table). Silica (Si) dynamics in *A. kenti* were not influenced by treatment, but declined consistently over time (S2 Fig, S7 Table). In contrast, Si concentrations for *A. spathulata* increased throughout the experiment, with minimal differences among treatments (S2 Fig, S7 Table).

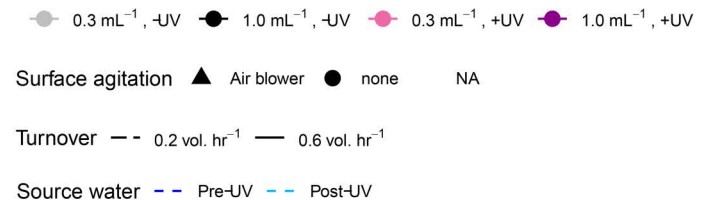

**A**

**B**

**C**

**D**

**E**

**Fig 3. Selected nitrogen water quality parameters and bacterial abundance for *A. spathulata* and *A. kenti*** in different culture treatments: A) nitrite and ($NO_2^-$; µM) B) nitrate ($NO_3^-$; µM), and C) particulate nitrogen (µM); D) particulate carbon (µM) and E) bacterial abundance measured from a subset of treatments using ddPCR (cells mL$^{-1}$). The gray band in E indicates the mean ± SE of bacterial abundance from local reef samples.

Columns distinguish coral species. Larval stocking densities and UV sterilization are represented using different colors, tank turnover treatments have solid or dashed lines, and the surface agitation treatment is denoted with triangle symbols. Points represent means and error bars represent SE. Blue lines represent values in the incoming seawater before (dark blue) and after UV sterilization (light blue). The black vertical line indicates when zygotes were added.

**Table 3. Linear model results for water quality characteristics of larval cultures.**

| | | Day in culture | | | Treatment | | | Treatment * Day | | |
|---|---|---|---|---|---|---|---|---|---|---|
| | Response | Num. df | F | p | Num. df | F | p | Num. df | F | p |
| *A. kenti* | Temp. | 5 | 36.9 | **<0.01** | 5 | 11.4 | **<0.01** | 25 | 1.3 | 0.17 |
| | DO | 5 | 199.5 | **<0.01** | 5 | 3.6 | **0.03** | 25 | 5.1 | **<0.01** |
| | pH | 5 | 33.0 | **<0.01** | 5 | 3.9 | **0.03** | 25 | 1.8 | **0.03** |
| | TDN | 4 | 37.2 | **<0.01** | 5 | 1.8 | 0.13 | 20 | 2.6 | **<0.01** |
| | $NH_4^+$ | 4 | 33.7 | **<0.01** | 5 | 0.4 | 0.86 | 20 | 1.9 | **0.02** |
| | $NO_2^-$ | 4 | 225.1 | **<0.01** | 5 | 12.2 | **<0.01** | 20 | 25.1 | **<0.01** |
| | $NO_3^-$ | 4 | 612.1 | **<0.01** | 5 | 99.1 | **<0.01** | 20 | 36.9 | **<0.01** |
| | DOC | 4 | 348.2 | **<0.01** | 5 | 5.0 | **0.01** | 20 | 10.8 | **<0.01** |
| | Diss. P | 4 | 116.7 | **<0.01** | 5 | 27.6 | **<0.01** | 20 | 6.2 | **<0.01** |
| | Si | 4 | 413.2 | **<0.01** | 5 | 2.4 | **0.04** | 20 | 1.1 | 0.39 |
| | Part. C | 4 | 229.1 | **<0.01** | 5 | 2.9 | 0.06 | 20 | 2.9 | **<0.01** |
| | Part. N | 4 | 276.4 | **<0.01** | 5 | 9.0 | **<0.01** | 20 | 3.2 | **<0.01** |
| *A. spathulata* | Temp. | 5 | 43.6 | **<0.01** | 5 | 4.5 | **0.01** | 33 | 2.1 | **<0.01** |
| | DO | 5 | 61.9 | **<0.01** | 5 | 1.3 | 0.34 | 33 | 2.0 | **<0.01** |
| | pH | 5 | 78.7 | **<0.01** | 5 | 11.8 | **<0.01** | 33 | 5.6 | **<0.01** |
| | TDN | 5 | 25.0 | **<0.01** | 5 | 4.4 | **0.01** | 21 | 2.7 | **<0.01** |
| | $NH_4^+$ | 5 | 16.6 | **<0.01** | 5 | 4.6 | **<0.01** | 21 | 2.4 | **<0.01** |
| | $NO_2^-$ | 5 | 726.5 | **<0.01** | 5 | 4.0 | **0.02** | 21 | 4.7 | **<0.01** |
| | $NO_3^-$ | 5 | 241.7 | **<0.01** | 5 | 24.4 | **<0.01** | 21 | 31.3 | **<0.01** |
| | DOC | 5 | 47.4 | **<0.01** | 5 | 1.1 | 0.41 | 21 | 4.4 | **<0.01** |
| | Diss. P | 5 | 17.2 | **<0.01** | 5 | 1.7 | 0.22 | 21 | 5.0 | **<0.01** |
| | Si | 5 | 3124.0 | **<0.01** | 5 | 5.3 | **0.01** | 21 | 4.3 | **<0.01** |
| | Part. C | 5 | 346.8 | **<0.01** | 5 | 22.4 | **<0.01** | 20 | 29.3 | **<0.01** |
| | Part. N | 5 | 407.6 | **<0.01** | 5 | 8.0 | **<0.01** | 20 | 8.1 | **<0.01** |

Data were analyzed using linear mixed models using the day in culture, culture treatment, and their interaction as fixed predictors. The table includes the degrees of freedom (df), F statistic (F), and p value. P-values ≤0.05 are indicated in bold.

Temperature, dissolved oxygen (DO), and pH remained stable throughout the larval rearing period and remained within expected ranges for tropical seawater, although minor differences were detected (Table 3, S3 Fig, S3 Table, S5 Table, S6 Table, S7 Table).

## Microbial community of culture water

The total bacterial and *Vibrio* abundances in seawater for both *A. kenti* and *A. spathulata* larval tanks varied significantly over time depending on culture treatment (treatment*Day; Table 2, S4 Table). However, there were no consistent treatment effects on microbial abundance across both species (Fig 3E). In *A. kenti*, total bacterial abundance was consistently higher in cultures with 0.2 vol. hr$^{-1}$ rate of turnover compared to 0.6 vol. hr$^{-1}$, although this difference was already evident prior to larval stocking and was not observed for *A. spathulata* (S4 Table).

In *A. kenti,* bacterial abundance in 0.2 vol. hr⁻¹ cultures reached five times higher than the source seawater (Fig 3E). Despite this increase, there was no associated increase in larval mortality or malformations (Fig 2B). All other treatments, including those for *A. spathulata*, showed bacterial and *Vibrio* abundances comparable to the source seawater (Fig 3E, S5 Fig.). Moreover, the culture systems and source seawater contained ~75% of the total bacterial abundance observed in local reef water and supported relatively low *Vibrio* levels, indicating that the culture systems were relatively hygienic (Fig 3E, S5 Fig.). For both species, culture treatments had no clear or consistent effect on *Vibrio* spp. abundance (S4 Table).

Microbial community diversity, as measured by Faith's phylogenetic diversity index, was higher in tanks with higher turnover rates, significantly so for *A. kenti*, where diversity increased by 72%, but not significantly for *A. spathulata* which showed only a 34% increase (Fig 4A, Table 2). Phylogenetic diversity also changed significantly over time for both species, although in opposite directions: it generally increased for *A. kenti* and decreased for *A. spathulata* (Fig 4A). In *A. kenti*, diversity declined significantly on Day 1 but increased on Days 2 and 4, whereas diversity in *A. spathulata* exhibited a significant increase only on Day 2 (S3 Table). Community composition was relatively stable across Days 1–2 for both species; however, only *A. spathulata* exhibited significant temporal shifts in microbial composition across the full culture period (Fig 4B, Table 4). Overall, microbial phylogenetic diversity in larval cultures was lower than in the source seawater (Fig 4A).

Bacterial composition in larval cultures was dominated by *Alphaproteobacteria* (19% in *A. kenti* and 26% in *A. spathulata*) and *Gammaproteobacteria* (73% in *A. kenti* and 71% in *A. spathulata*). *Alphaproteobacteria* predominately belonged to the genera *Citreicella*, *Cognatishimia,* and *Donghicola* while *Gammaproteobacteria* predominantly belonged to the genera *Aestuariibacter*, *Alteromonas*, *Marinicella*, *Neptuniibacter*, *Oleibacter*, *Pseudomonas*, and an unidentified genus in the family *Saccharaospirillaceae* (Fig 5). In contrast, the source seawater exhibited higher phylogenetic diversity (Fig 4A), with 44% *Alphaproteobacteria*, 25% *Gammaproteobacteria*, and greater representation of *Planctomycetota*, *Marinimicrobia*, and *Verrucomicrobiota* than the larval cultures.

Water turnover significantly influenced the microbial community composition in both species (Fig 4B, Table 4). In *A. kenti* cultures, high turnover was associated with decreased relative abundances of *Gammaproteobacteria* (Fig 5), which comprised 20 of the 24 significantly different ASV between turnover treatments (S6 Fig.). Seven significant *Neptuniibacter* and 2 *Alteromonas* ASV decreased in abundance (from ~10–2% each) while 8 *Oleibacter* ASV increased (from ~1 to –3% each) in the 0.6 vol. hr⁻¹ *A. kenti* cultures (S6 Fig.). In *A. spathulata* cultures, turnover was associated with significant but smaller effects on community composition than in *A. kenti* (Fig 4, Fig 5, Table 4). No ASV were significantly different between 0.2 and 0.6 vol. hr⁻¹ *A. spathulata* cultures.

Bacterial composition also varied significantly over time but only for *A. spathulata* (Fig 4B, Table 4). Changes over time were predominantly seen in *Gammaproteobacterial* ASV, which comprised 64 of the 76 significantly different ASV between Days 1 and 6 (S7 Fig.). The 76 significant ASV were evenly distributed amongst 21 genera, with the clearest differences being decreases in 5 *Alteromonas* ASV (by ≤10% rel. abund. each), increases in 5 *Oleibacter* ASV (by ≤4% each), and increases in 4 unclassified ASV in the family Saccharospirillaceae by Day 6 (by ≤10% each; S7 Fig.). On the other hand, 4 significant *Donghicola* ASV appeared to peak in abundance on Days 2 and 3, begin to decrease, and have lower abundance on Day 6 compared to Day 1 (by ≤1% each; S7 Fig.).

## Discussion

Restoration programs will increasingly be reliant on sexual propagation of corals to generate large numbers of healthy larvae to supply reef interventions [5,11,43]. Although current larval production meets research and development needs, improving efficiency and scale across all developmental stages will become critical as restoration efforts expand [5]. Coral embryogenesis is a sensitive and delicate process, which can be affected by temperature [16], water quality [44], and interactions with bacteria or macroalgae [19], highlighting the need for further research into the health and survival dynamics of larvae in larger-scale rearing scenarios. In this study, we investigated factors that were hypothesized to influence

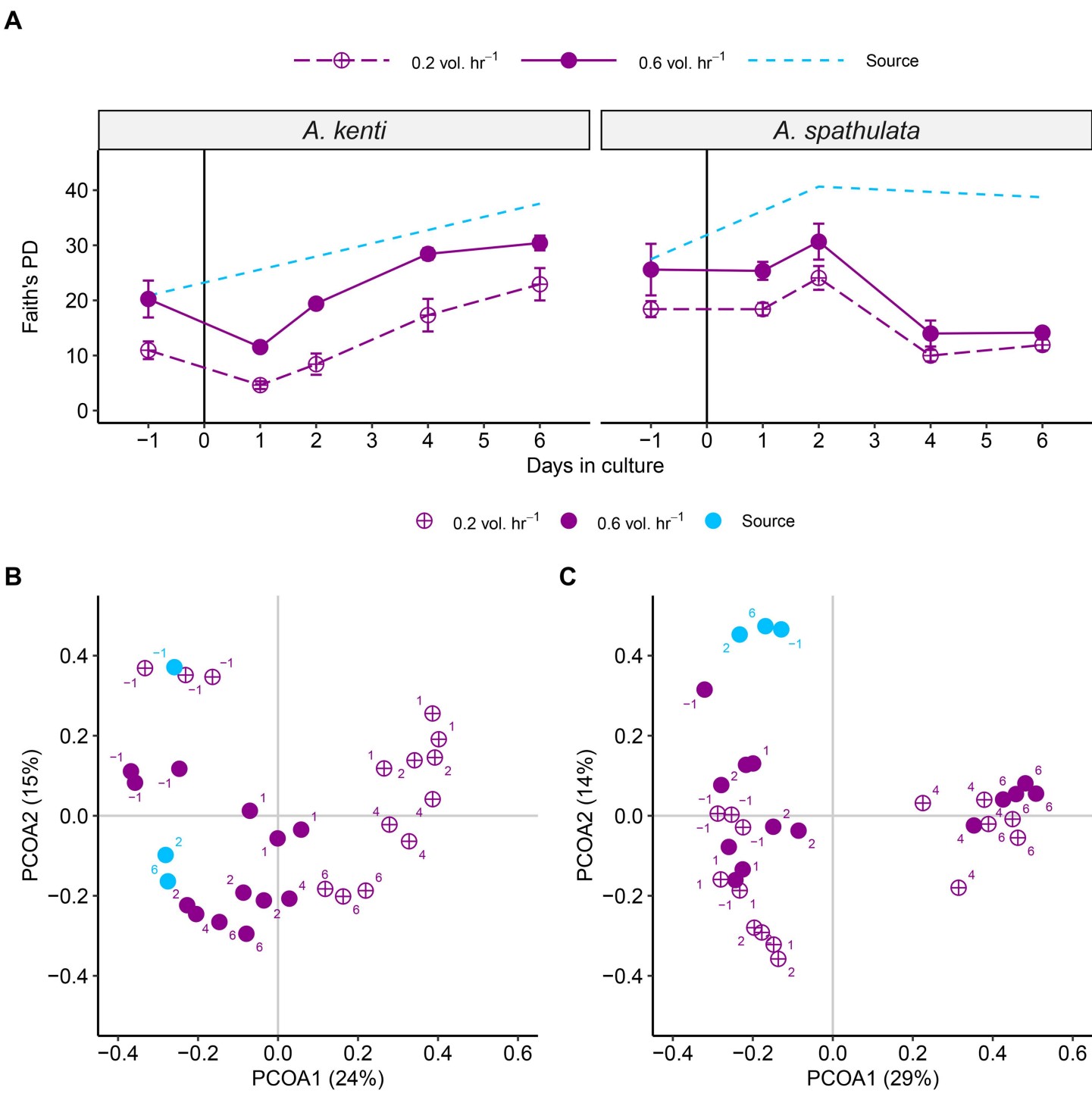

**Fig 4. Dynamics of microbial community diversity and composition in culture water.** A) Faith's phylogenetic diversity (PD) of the bacterial community from 16S rDNA sequencing. The culture treatments are indicated in purple while the source seawater is indicated in blue. The low turnover (0.2 vol. hr$^{-1}$) treatment is indicated using purple dashed lines while the high turnover (0.6 vol. hr$^{-1}$) treatments is indicated with purple solid lines. Points represent means and error bars represent SE. The vertical black line represents when larvae were added to the culture tanks. B and C) Ordinations by principal coordinates analysis of the microbial communities between 0.2 and 0.6 × hr$^{-1}$ turnover treatments for *A. spathulata* and *A. kenti*, respectively. Point locations represent the relative community composition of a particular culture treatment on each day of culture (indicated using numbers). Low turnover is indicated using open purple circles and high turnover is indicated using solid purple symbols. Blue asterisks indicate the community composition of the source seawater. X and Y axes are the first and second principal coordinates, respectively.

**Table 4. Permutational analysis of variance (PERMANOVA) results of microbial communities in *A. kenti* and *A. spathulata* larval cultures. Predictors include the day of culture, water turnover (0.2 or 0.6 vol. hr⁻¹) and their interaction. P-values ≤0.05 are indicated in bold.**

| | Day in culture | | | | Turnover | | | | Turnover * Day | | | |
|---|---|---|---|---|---|---|---|---|---|---|---|---|
| | df | F | p | $R^2$ | df | F | p | $R^2$ | df | F | p | $R^2$ |
| *A. kenti* | 4 | 0.6 | 0.99 | 0.09 | 1 | 2.8 | **0.01** | 0.11 | 4 | 0.6 | 0.97 | 0.10 |
| *A. spathulata* | 4 | 5.5 | **<0.01** | 0.45 | 1 | 2.7 | **0.01** | 0.06 | 4 | 1.2 | 0.18 | 0.10 |

larval survival and health, including stocking density, tank turnover rates, water sterilization, and water agitation. Across treatments, > 40% larval survival was achieved with turnover rates as low as 0.2 vol. hr⁻¹ and densities up to 1 larvae mL⁻¹. Species-specific responses were evident: *A. kenti* larvae maintained high survival across all culture conditions, whereas *A. spathulata* larvae showed a consistent decline in survival over time.

Coral breeding of individual species offers advantages over capturing and rearing multi-species coral spawn slicks in the field. While fertilization rates are similar (~86%; [15]), larval survival was substantially higher (i.e., 40–100% for *A. spathulata* and *A. kenti*, respectively) compared to ~13% in natural slicks over the same developmental period [15,45]. For single-species breeding, the benefits of controlled ex situ larval rearing were confirmed by Randall et al. [22], whereby a cohort of *A. millepora* larvae had 17 × higher survival when cultured in vessel rearing systems compared to in situ larval rearing pools. Furthermore, settlement efficiency in ex situ aquaculture can range between 44 and 81% (this study) compared to 3% observed for larvae from multi-species spawn slicks [15]. However, this discrepancy is likely more related to how the larvae are exposed to the settlement substrata than differences in larval quality [22]. In this study, the different culture treatments could have influenced the time required to reach settlement competency but will require further testing to evaluate. In general, ex situ methods improved settlement efficiency via 1) the use of species-specific settlement cues (e.g., specific CCA species; [28], [46]) and 2) settlement can be targeted for peak larval competency for each coral species [27]. These results highlight the potential of ex situ approaches to improve coral production, although costs need to be reduced to make it a viable option compared to in situ methods [22].

Stocking density is an important factor for improving larval production efficiency. In this study, we found no consistent differences in larval survival, size, appearance, or settlement success between cultures stocked at 0.3 and 1.0 larvae mL⁻¹ in 500 L flow-through tanks. Although higher densities (>1.0 larvae mL⁻¹) have been used successfully in small-scale systems, buoyant developing zygotes can experience increased mortality risks due to oxygen depletion from surface crowding and waste accumulation. Most studies report successful culturing at densities <1 larvae mL⁻¹ [8,47,48], though densities up to 10 larvae mL⁻¹ have been achieved when excess sperm and unfertilized eggs have been removed early [49]. Our findings indicate densities of 1.0 larvae mL⁻¹ are feasible in large-volume flow-through systems and may offer a practical upper threshold for routine operations, with potential for further optimization. However, attempts to culture at 2.0 larvae mL⁻¹ were unsuccessful due to larvae clogging filters (and thus the culture overflowing), likely caused by the combination of high stocking density and tank turnover rates. This highlights the need for system modifications, such as enhanced filtration capacity or alternate outlet designs, to support higher stocking densities in large-scale applications.

Water turnover is essential for managing waste and oxygen demand in aquaculture systems [50]. We evaluated turnover rates of 0.2 and 0.6 vol. hr⁻¹ in flow-through systems, both higher than the ≤ 0.1 vol. hr⁻¹ rates typically used in coral larval production. While the 0.2 vol. hr⁻¹ treatment was associated with elevated particulate C and N, particularly in higher-density cultures (Figs 3C, 3D), larval survival and settlement success were comparable to those observed at 0.6 vol. hr⁻¹. However, the higher turnover rate (0.6 vol. hr⁻¹) appeared to increase the incidence of malformed *A. spathulata* larvae (Fig 2D), potentially due to greater turbulence or mechanical stress at the cylindrical outflow mesh filter. These results suggest that moderate turnover rates, such as 0.2 vol. hr⁻¹, may strike the balance between maintaining water quality and minimizing larval damage, while also reducing water usage.

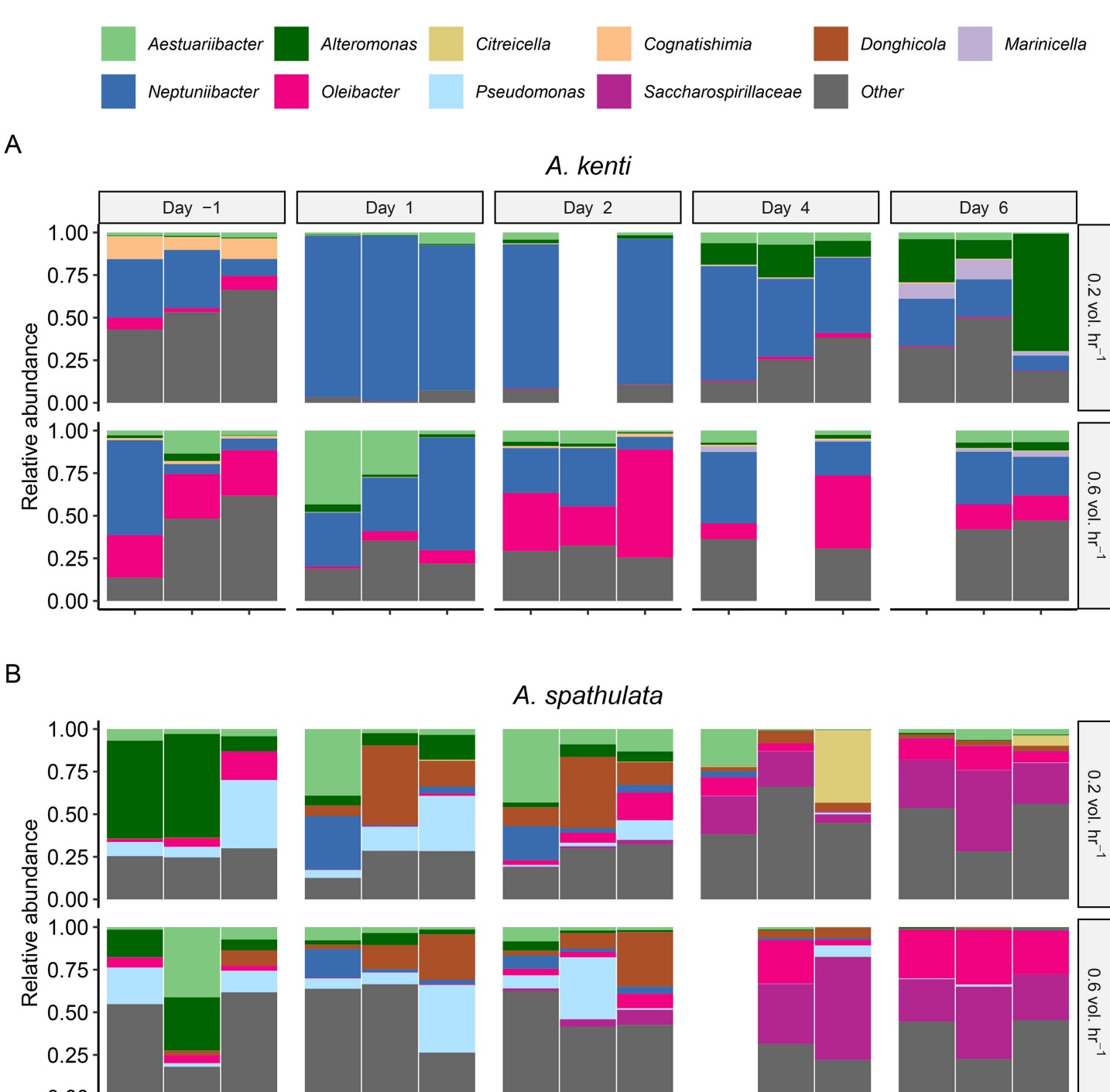

**Fig 5. Genus-level microbial community composition in the culture water for *A. kenti* (A) and *A. spathulata* (B).** Every panel depicts the composition of up to 3 culture tanks for each sampling Day and turnover treatment. Each bar represent relative abundance of genera that exceeded 10% relative abundance in any coral species, turnover treatment, or sampling day. *Citreicella*, *Cognatishimia*, and *Donghicola* belong the class *Alphaproteobacteria* while the other genera belong to the *Gammaproteobacteria*. 'Other' includes all other genera as well as three unidentified taxa with >10% relative abundance but that could not be identified to genus-level.

Both water quality and environmental microbial dynamics are known contributors to larval mortality [5,18]. Although stocking density did not directly affect larval quality or survival in our study, it appeared to influence water quality. Ammonia concentrations–known to impair fertilization at higher levels (>200 µmol L$^{-1}$, [51]) –remained low (<2.5 µmol L$^{-1}$) and were comparable to incoming seawater. Interestingly, nitrite ($NO_2^-$) and nitrate ($NO_3^-$) concentrations were depleted in high-density cultures, potentially due to microbial denitrification [52]. The concurrent increase in particulate N and C, along with reductions in $NO_3^-$ in both species, support the hypothesis that microbial activity contributed to the depletion of dissolved nitrogen species (but not total diss. N). Notably, UV sterilization of the incoming water had no effect on $NO_2^-$ and $NO_3^-$ levels, suggesting that in-tank microbial processes, or microbial communities associated with the developing embryos, may have driven these observed changes in nitrogen dynamics.

Microbial monitoring proved to be a sensitive tool for tracking conditions in coral larval cultures, even in the absence of overt signs of culture failure. Overall, microbial loads in the culture water were low compared to environmental samples from reef habitats. For each coral species, the microbial communities experienced consistent changes among replicate cultures. A temporary increase in bacterial abundance was detected in low-turnover *A. kenti* cultures, coinciding with depleted dissolved $NO_2^-$ and $NO_3^-$. Notably, *Vibrio* spp., a group commonly associated with coral diseases [19–21], were present at 10-fold lower abundances in culture tanks than in reef seawater, and were not associated with the low survivorship of *A. spathulata* larvae. In contrast, bacteria from the family *Saccharospirillaceae* increased in relative abundance as *A. spathulata* cultures declined, indicating a rapid microbial response. These microbial shifts, including the high bacterial load on days 1 and 2 in low-turnover *A. kenti* cultures and community composition changes on days 4 and 6 in *A. spathulata* cultures, demonstrate that microbial communities are responsive to culture conditions. These findings suggest that integrating microbial and nutrient monitoring into larval culture protocols could detect early signs of culture instability and improve our understanding of culture collapse.

The 16S rRNA gene analysis revealed that bacterial communities in larval cultures had lower phylogenetic diversity than those in incoming seawater. Both tank turnover rates and larval mortality coincided with differences in specific bacterial taxa, particularly genera within the *Gammaproteobacteria*, many of which are known nitrate reducers [53]. In *A. kenti* cultures, the *Neptuniibacter* and *Alteromonas* increased in relative abundance during and after nutrient depletion, suggesting their potential roles in nutrient cycling. These genera are also associated with coral reproductive processes: *Neptuniibacter* has been linked to larval settlement [54], while *Alteromonas* has been detected during the gamete release [55] and is a strong nitrate competitor [56]. These findings suggest that bacteria introduced with the zygotes may influence subsequent larval culture dynamics. Importantly, attempts to sterilize cultures using antibiotics have previously led to increased larval mortality and decreased settlement rates [19], suggesting that some bacteria may play beneficial roles in larval development.

Despite extensive monitoring, the causes of larval mortality remain poorly understood. *A. spathulata* larvae exhibited a gradual decline in survival across our experimental treatments, potentially due to cross-fertilization between incompatible genotypes [57], or environmental factors impacting gamete quality (e.g., temperature; [58]). High mortality rates of *A. spathulata* larvae have been reported elsewhere [59], potentially due to the relatively high metabolic demand of larvae up to when they become competent to metamorphose [14]. In contrast, *A. kenti* had consistently high survival under the same conditions, suggesting that even closely-related species experience different mortality rates during the larval phase.

This study shows that large-volume, flow-through aquaculture systems can effectively rear coral larvae at densities up to 1.0 larvae mL$^{-1}$, offering a scalable solution for restoration programs. *Acropora kenti* maintained high survival across treatments, while *A. spathulata* showed species-specific declines, likely due to intrinsic traits rather than culture conditions. Culture parameters such as turnover rate (up to 0.6 vol. h$^{-1}$), UV treatment, and agitation had minimal consistent effects on larval health, indicating that corals can tolerate a range of conditions if water quality is maintained. Microbial monitoring revealed that water turnover and larval mortality influenced bacterial communities, with some potentially beneficial taxa (e.g., *Alteromonas, Neptuniibacter*) increasing during nutrient depletion. These

shifts occurred without elevated pathogenic *Vibrio* spp., suggesting that natural microbial communities in flow-through systems may support larval health. Future research should focus on improving system design for higher densities, integrating microbial diagnostics, and tailoring protocols to species-specific needs to enhance aquaculture efficiency and restoration outcomes.

## Supporting information

**S1 Fig. Daily temperature (°C), dissolved oxygen (DO; % saturation), and pH for cultures of *Acropora kenti* and *Acropora spathulata* larvae.** Points represent means of culture tanks from each treatment. Triangular points distinguish tanks with surface agitation while dashed lines distinguish tanks at higher turnover. Gray or black points represent tanks without UV sterilization while pink or purple points represent tanks with UV sterilization, with darker shades (black and purple) representing tanks with at higher larval density. Blue dashed lines represent values in the incoming seawater before (dark blue) and after UV sterilization (light blue). The black vertical line distinguishes samples taken before and after larvae were added.
(DOCX)

**S2 Fig. Example *A. spathulata* larvae that are abnormal or normal (elongated, or round).** Larvae are approximately $0.06\,mm^2$.
(DOCX)

**S3 Fig. Larval settlement proportion for *A. kenti* and *A. spathulata* 5 days post-fertilization.** The x-axis is water volume turnover (vol. hr$^{-1}$) and the y-axis is the proportion of larvae settled. Colors distinguish the stocking density (light vs dark) and UV sterilization (shades of purple versus black) culture treatments. For *A. kenti*, surface agitation is shown using a triangle.
(DOCX)

**S4 Fig. Nutrient levels in culture tanks including total dissolved N (TDN; μM), ammonium (NH$_4^+$), dissolved organic carbon (DOC, mg L$^{-1}$), total dissolved phosphorus (TDP; μmol L$^{-1}$), and silica (Si; μmol L$^{-1}$).** Columns distinguish coral species. Larval stocking densities are represented using different colors, tank turnover treatments have solid or dashed lines, and the surface agitation treatment is denoted with triangles. Points represent means and error bars represent SE. Blue lines represent values in the incoming seawater before (dark blue) and after UV sterilization (light blue). The black vertical line distinguishes samples taken before and after larvae were added.
(DOCX)

**S5 Fig. *Vibrio* spp. abundance (cells mL$^{-1}$) measured using ddPCR.** Points and error bars represent mean and SE, respectively. The gray band in D indicates the mean±SE of bacterial abundance from local reef samples. Columns distinguish coral species. Larval stocking densities and UV sterilization are represented using different colors and tank turnover treatments have solid or dashed lines. The blue line represents values in the incoming after UV sterilization. The black vertical line indicates when larvae were added.
(DOCX)

**S6 Fig. Relative abundance of 24 ASV in *Acropora kenti* cultures with significantly different abundance between 0.2 and 0.6 vol. hr$^{-1}$ treatments.** ASV abundances are grouped by the turnover treatments and genera. Points represent mean abundance for each ASV and error bars represent SE.
(DOCX)

**S7 Fig. Relative abundance of 76 ASV in *Acropora spathulata* cultures with significantly different abundance between Day 1 and Day 6 of culture.** ASV abundances are grouped by Phylum:Genus (ASV are unclassified where

genus is not provided). The blue points and lines represent the abundance of the same ASV in the source water. Points represent mean abundance for each ASV and error bars represent SE.
(DOCX)

**S1 Table. Sampling schedule for different responses.**
(DOCX)

**S2 Table. Description of post hoc comparisons.** For responses with significant treatment*day interactions, all treatments were compared using Tukey post hoc comparisons. From these comparisons, we interpret the results of 6 pairwise comparisons for *Acropora kenti* and 5 for *Acropora spathulata* that differed in a single culture treatment condition (Tested treatment). Two comparisons are used to test differences due to stocking density and sterilization and are denoted using superscript [ab] and [cd], respectively. The controlled treatments represent the shared culture conditions for each comparison.
(DOCX)

**S3 Table. Significant post hoc comparisons for main effects of time (Table 2, Table 3) for larval appearance, water temperature, silica (Si) concentrations, and Faith's phylogenetic diversity.** The table includes the response variables and the significant differences between consecutive days (e.g., Day 1 − Day −1, Day 2 − Day 1, etc). Each cell contains an effect size (difference) and an associated statistical significance (p value). Empty cells indicate pairwise comparisons with $p > 0.05$ while dashes indicate days where a given response was not measured. Responses without significant differences on any day are not shown. * this difference in Si was measured on Day 6 − Day 4.
(DOCX)

**S4 Table. Significant post hoc comparisons for treatment*time interactions on larval, bacterial, and *Vibrio* responses (Table 2).** The table reports the species, the measured response, treatments being compared, and the significant effects on each experimental day. Each cell contains an effect size (difference or ratio) and statistical significance (p value). Empty cells indicate pairwise comparisons with $p > 0.05$ while dashes indicate days where a given response was not measured. Responses without significant differences on any day are not shown. [b] refers to the comparison of stocking densities in tanks with UV sterilization in S2 Table.
(DOCX)

**S5 Table. Significant pairwise comparisons for main effects of treatment for larval appearance, larval settlement, temperature, and Faith's phylogenetic diversity.** The contrast indicates the effect size (ratio or difference) and statistical significance (*p* value) for each comparison. [c] and [d] refer to the comparisons of sterilization at 0.3 and 1.0 larvae mL$^{-1}$ in S2 Table, respectively.
(DOCX)

**S6 Table. Significant post hoc comparisons of water quality between culture treatments for *A. kenti* (Treatment*Day and treatment main effects).** Responses include nitrite ($NO_2^-$), nitrate ($NO_3^-$), dissolved organic carbon (DOC), particulate carbon (Part. C), dissolved oxygen (DO), pH, total dissolved phosphorous (TDP), and particulate nitrogen (Part. N). The table includes the independent and dependent variables and the significant comparisons on each experimental day. Empty cells indicate pairwise comparisons with $p > 0.05$. Pairwise comparisons between treatments that differed in multiple parameters are not reported. Superscript are used to distinguish between tank conditions for multiple pairwise comparisons: [a] compare stocking densities without UV, [b] compare stocking densities with UV, [c] compare sterilization at 0.3 larvae mL$^{-1}$, and [d] compare sterilization at 1.0 larvae mL$^{-1}$.
(DOCX)

**S7 Table. Significant post hoc comparisons of water quality between culture treatments for *A. spathulata* (Treatment*Day and treatment main effects).** Responses include nitrite ($NO_2^-$), nitrate ($NO_3^-$), dissolved organic carbon (DOC),

 

particulate carbon (Part. C), dissolved oxygen (DO), pH, total dissolved phosphorous (TDP), and particulate nitrogen (Part. N). The table includes the independent and dependent variables and the significant comparisons on each experimental day. Empty cells indicate pairwise comparisons with $p > 0.05$. Pairwise comparisons between treatments that differed in multiple parameters are not reported. Superscript are used to distinguish between tank conditions for multiple pairwise comparisons: [a] compare stocking densities without UV, [b] compare stocking densities with UV, [c] compare sterilization at 0.3 larvae mL$^{-1}$, and [d] compare sterilization at 1.0 larvae mL$^{-1}$.
(DOCX)

## Acknowledgments

We thank F. Flores for their assistance conducting experiment. In addition, we thank the AIMS National Sea Simulator staff, especially G. Milton and A. Marc for their assistance with construction and testing of the aquarium system. We acknowledge the Bindal and Manbarra People as the Traditional Owners where this work took place. We pay our respects to their Elders past, present and emerging and we acknowledge their continuing spiritual connection to their land and sea country.

## Author contributions

**Conceptualization:** David G. Bourne, Andrew P. Negri, Andrea Severati, Muhammad Azmi Abdul Wahab.

**Data curation:** Sophie Ferguson, Yilmaz Can Hiçyilmaz.

**Formal analysis:** Blake D Ramsby.

**Investigation:** Ramona Brunner, Jonathan Barton, Sophie Ferguson, Clare Grimm, Yilmaz Can Hiçyilmaz, Andrew P. Negri, Yui Sato, Muhammad Azmi Abdul Wahab.

**Methodology:** Ramona Brunner, Jonathan Barton, Sophie Ferguson, Clare Grimm, Yilmaz Can Hiçyilmaz, Andrew P. Negri, Yui Sato, Muhammad Azmi Abdul Wahab.

**Project administration:** Muhammad Azmi Abdul Wahab.

**Resources:** Andrea Severati.

**Supervision:** David G. Bourne, Yui Sato, Muhammad Azmi Abdul Wahab.

**Visualization:** Blake D Ramsby, Ramona Brunner, Yilmaz Can Hiçyilmaz.

**Writing – original draft:** Blake D Ramsby.

**Writing – review & editing:** Blake D Ramsby, Ramona Brunner, Sophie Ferguson, Clare Grimm, David G. Bourne, Andrew P. Negri, Yui Sato, Muhammad Azmi Abdul Wahab.

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
