## [Decision Letter · Decision Letter 0]

29 Sep 2025

Dear Dr. Ramsby,

Thank you for submitting your manuscript to PLOS ONE. After careful consideration, we feel that it has merit but does not fully meet PLOS ONE’s publication criteria as it currently stands. Therefore, we invite you to submit a revised version of the manuscript that addresses the points raised during the review process.

We look forward to receiving your revised manuscript.

Kind regards,

Satheesh Sathianeson, Ph.D

Academic Editor

PLOS ONE

Journal Requirements:

2. Thank you for stating the following financial disclosure: [This research was supported by the Reef Restoration and Adaptation Program (RRAP), which is funded by a partnership between the Australian Government’s Reef Trust and the Great Barrier Reef Foundation.].

Reviewers' comments:

Reviewer's Responses to Questions

**Comments to the Author**

1. Is the manuscript technically sound, and do the data support the conclusions?

Reviewer #1: No

Reviewer #2: Yes

Reviewer #3: Yes

2. Has the statistical analysis been performed appropriately and rigorously?

Reviewer #1: Yes

Reviewer #2: Yes

Reviewer #3: Yes

3. Have the authors made all data underlying the findings in their manuscript fully available?

Reviewer #1: Yes

Reviewer #2: Yes

Reviewer #3: Yes

4. Is the manuscript presented in an intelligible fashion and written in standard English?

Reviewer #1: Yes

Reviewer #2: Yes

Reviewer #3: Yes

Reviewer #1: This paper describes a multi-factorial experimental study evaluating the performance of coral larvae, water quality and microbial communities in an at-scale aquaculture setting. The study addressed multiple culture treatment conditions that would be expected to affect larval performance, including water turnover rate, UV sterilization, water surface agitation, and larval stocking density. Surprisingly, the results reported show none of these different culture conditions significantly affected larval survival, visual condition, nor settlement competence within the ranges tested, though I have some concerns with how these results are reported ( or not reported) which are articulated below. The results do strongly emphasize intrinsic factors in coral larval success as the two species examined had radically different performance across all the experimental culture treatments. Meanwhile, microbial communities and some water parameters did respond to the treatments. Substantive concerns include:

- Ln. 110-111: Fertilization succcess is never mentioned; only that 80% of eggs were cleaving at 1.5 hr after crossing. Does this indicate that the fertilization rate was 80% or is the assumption that all were fertilized and 20% simply did not manifest cleavage yet? And was this number identical for both species in the two different months (seems unlikely)? This is a fairly important distinction (ie. whether there were 20% of unfertilized eggs incorporated (and therefore adding dead biomass) in the cultures or not) and thus fertilization needs to be reported.

- Ln 171-175: This paragraph states events that are dismissed and not given as results, but perhaps they should be (or at the very least such dismissal needs to be clearly justified)? Four out of 18 tanks of A.kenti across three treatments essentially failed due to ‘loss of larvae through filters’. What does this mean exactly and what is the justification for dismissing that the majority of replicates of the ‘air surface agitation treatment’ failed in this way?

And more importantly, one of the treatment combinations for A.spathulata failed completely (the 2 larvae ml-l x .6 turnover). This latter point seems like a (somewhat important) result (that could provide important guidance to future efforts) but it is essentially dismissed and not even reported as a result. Statements in the results such as ‘For both species, survival was not affected by culture treatments’ do not seem strictly true given that this treatment combination failed.

- Ln 187-188: Under the condition scoring, it is acknowledged in the following sentence that both ‘round’ and ‘elongated’ larval shapes are appropriate. These different shapes occur because oval larvae can be oriented either parallel or perpendicular to the plane of view in any given image. But this means a single larvae would have substantially different sizes if it was imaged in a parallel (i.e. elongate) versus in a perpendicular (i.e. round) view. Was this controlled for in the larval size measurements (i.e. were only ‘round’ or only ‘elongate’ larvae used for size measurements)?

- Fig 2: The authors mention that noise in the density measurements can account for survivorship estimates over 100%. However, I am struck with the degree of this overestimation for A.kenti (essentially all replicates and time points estimate > 100% survivorship, suggesting that the initial estimate of stocking density was perhaps substantial overestimated (as much as 50% for the kenti, surface agitation treatment which does appear to decline substantially over time (~ 150% to ~ 100%), contrary to the statistical conclusions, but I guess this is also the treatment that had only one replicate . . .) and the difference in the degree of this artifact between species. In short, the discrepancy of the A.kenti data raise concerns about the degree of artifact in (and therefore reliability) the larval density measurements.

- Settlement assays were conducted only once and so any differences in the observed settlement (i.e., between either species or between treatments) could be due to variation in the timing of competence rather than indicating an overall lack of fitness (as is implied). This should be acknowledged in the discussion

A few needed details and other suggestions are included below.

- Ln 478: ‘investigated factors that influence’ probably should be re-phrased as ‘factors that were hypothesized to influence’ (since the prevailing conclusion is that they did not)

- Ln 492: improve should be improved. Also, this discussion confounds ‘in situ’ culture with the use of mixed-species slicks. In situ larval culture can also be conducted from single-species gamete collections (e.g., Miller et al. 2022. Rest Ecol 30: e13512 . doi: 10.1111/rec.13512).

- Ln 501: zygote should be zygotes

- Starting Ln 551: reference formatting switches here for a paragraph or two (and I believe the refs in this section do not appear in the list)

Reviewer #2: Review summary

Here the authors present a study investigating the effects of modifying various culture conditions on larval rearing success for two Acropora corals. Overall, the experiments and data analysis are well-designed and scientifically robust, the manuscript is highly cogent and thorough, and the results have important implications for coral larval aquaculture, making them worthy of publication. I have a few minor comments, but believe that the manuscript will be suitable for publication once these are addressed. Thank you to the authors for your important work and for making this an easy and interesting manuscript to review.

Specific comments

L1 — as soon as the title, and at other relevant places throughout the manuscript, the authors should clarify that the microbial communities analyzed here were from seawater samples, rather than coral-associated microbes. As of now, this is not clear until one gets to the relevant methods subsection, making some key features of the manuscript (namely, the title and abstract) somewhat confusing/unintentionally misleading.

L127–128 — using larvae from four colonies is sufficient but might mean that the results of the study do not perfectly translate to larger mixed-genotype cultures; this should be briefly mentioned in the discussion as an important caveat.

L185–193 — given how many larvae were in the cultures, I am surprised at the very small sample sizes for these metrics, and feel that this should also be mentioned in the discussion as an important caveat.

L203–204 — please provide information regarding how this instrument was calibrated and with what frequency.

L256 — please specify the range and average number of reads acquired per library.

L330–331 — fix reference.

Reviewer #3: The manuscript “Coral larval aquaculture: species specific survival and microbial dynamics in flow through systems” addresses important, yet overlooked, concepts in coral aquaculture. Although there are multiple successful methods of coral husbandry are well recorded and in practice worldwide, they are seldomly reported in the scientific literature and limited to word-of-mouth knowledge and protocols within grey literature. Therefore, more experiments that tests the efficiency of different methods are required to maximize the efficiency of ex situ nurseries. This study does a good job addressing these knowledge gaps by manipulating and testing the effects of multiple environmental factors on the survival and health of coral larvae. Most of my suggestions revolve around the description of their statistical analyses in the methods and that clarifying the language in some of their statements to tighten the accuracy, consistency, and rigorousness of the study. I recommend this manuscript for publication in PLoS ONE once these comments have been addressed. Please see my specific comments outlined below:

Introduction

Page 1, lines 49–50: “…poorly-defined…” is a strong statement that overstates the problem. Many coral nurseries have found ways to maximize their culture yields, but they are not published in peer-reviewed literature, as stated in Pollock et al. 2018. Instead, I suggest softening the language by acknowledging that maximizing larval yields has been addressed before but that there is still a need for rigorous data that determines what factors lead to maximizing those yields.

Page 2, lines 66–68: Where did these reported survival and stocking density values come from? The literature or from preliminary analyses? Please clarify or provide the appropriate references.

Methods

Page 6, lines 115–117: Was it 6.1 million per tank or in total for all 3 tanks? Please clarify.

Page 7, lines 137–138: Were these temperatures maintained using heaters? Please specify the equipment used to accomplish this.

Page 8, line 161: This reference is highlighted in gray and not in numeric form. Please fix it.

Page 10, line 214: Please change “was” to “were”

Page 12, line 261: Please write out what ASV stands for before using the acronym.

Page 12, lines 270–272: This sentence is hard to keep track of. So each response variable was tested against multiple predictors and time within the same model, and there was a model for each response variable, correct? I would prefer you list all of the predictor variables that were incorporated in the model (the “up to 6 listed” is also vague and uninformative, since there are more than six columns) and that from now on, “culture treatment” will be used as an umbrella term for all of those variables.

Page 12, line 274: If different error distributions were used, then they would be generalized linear models (GLMs) instead of linear models (LMs), which assume a Gaussian distribution and already have the error distribution built in . Please clarify the models that were used for the analyses.

Page 12, line 277: Please specify if the binomial GLM was done using relative abundance/percentage data or binary. Binary transformations can be quite a drastic transformation, so justification should be provided along with the criteria used to determine 1s and 0s if such a transformation was actually performed. If not, then please mention that relative abundance data were used for it.

Page 12, line 277–278: I assume that this was the situation for the Gaussian LM, but what diagnostic tests were used for the GLMs (i.e. normality and homogeneity of variance do not apply to GLMs with error distributions that are not Gaussian/normal)? Please clarify which diagnostics pertain to the LMs and which ones pertain to the GLMs.

Page 13, lines 296–297: Did you use the raw relative abundance data? If not, please specify whether the data were transformed/standardized in any way to reduce the influence of extreme values.

Results

Page 13, lines 303–305: If you state that there are species-level differences, please provide statistical tests that support it (i.e. a model that specifically tests variation among species). Otherwise, describe trends without implying significance. Also, please specify that these were the trends observed across all treatments.

Page 14, lines 330–331: “TableError! Reference source not found” please fix this issue.

Table 2 caption, line 334: You mention linear mixed models (LMEs) here, but in the main text you only refer to them as generic linear models. Please clarify which models exactly were used. If LMEs were used, please mention what you specified as your random effects in the models.

Page 16, lines 349–350: I recommend moving this statement (“Although differences in water quality were observed among culture treatments…”) to the beginning of the paragraph and merging it with the sentence that currently sits at the top of the paragraph (lines 348–349). This way, it leads more clearly into the rest of the paragraph than in the original draft.

Page 21, lines 436–437: Please specify which taxa in Fig. 5 belong to the classes mentioned here. I see that this is mentioned in the figure caption, but it should also be specified in the main text and in the figure legend so the reader doesn’t have to go digging for this information.

Discussion

Page 22, lines 474–475: Are you referring to mortality rates in the field, in the lab, or both? Please specify this and mention the hypothesized and/or observed causes of early life-stage mortality.

Page 23, line 492: “…facilitate improve…” choose one and remove the other.

Page 23, line 503: “…<1 larval mL-1…” and “…10 larval mL-1…” please change to larvae mL-1

Page 24, line 528: “with reductions of NO3–

Pages 24–15, lines 551, 555, 556, 559: Update these references to numerical format.

Fig. 1: Increase font size of the legend, I also recommend labelling the different larval stocking density clusters in the diagrams or adding in miniatures of the different cluster patterns in the legend to make it easier for readers to understand and follow the figure along.

**Do you want your identity to be public for this peer review?** For information about this choice, including consent withdrawal, please see our Privacy Policy

Reviewer #1: No

Reviewer #2: No

Reviewer #3: No

---

## [Author Response · Author response to Decision Letter 1]

19 Nov 2025

We thank the reviewers for the feedback and suggestions to improve this manuscript. Please find our responses in the attached revision.

---

## [Decision Letter · Decision Letter 1]

21 Dec 2025

Coral larval aquaculture: species-specific survival and microbial dynamics in flow-through systems

PONE-D-25-44441R1

Dear Dr. Ramsby,

We’re pleased to inform you that your manuscript has been judged scientifically suitable for publication and will be formally accepted for publication once it meets all outstanding technical requirements.

Kind regards,

Satheesh Sathianeson, Ph.D

Academic Editor

PLOS One

Additional Editor Comments (optional):

Reviewers' comments:

Reviewer's Responses to Questions

**Comments to the Author**

Reviewer #2: All comments have been addressed

Reviewer #3: All comments have been addressed

2. Is the manuscript technically sound, and do the data support the conclusions?

Reviewer #2: Yes

Reviewer #3: Yes

3. Has the statistical analysis been performed appropriately and rigorously?

Reviewer #2: Yes

Reviewer #3: Yes

4. Have the authors made all data underlying the findings in their manuscript fully available?

Reviewer #2: Yes

Reviewer #3: Yes

5. Is the manuscript presented in an intelligible fashion and written in standard English?

Reviewer #2: Yes

Reviewer #3: Yes

Reviewer #2: (No Response)

Reviewer #3: All my original comments have been adequately addressed and I recommend the manuscript for publication.

**Do you want your identity to be public for this peer review?** For information about this choice, including consent withdrawal, please see our Privacy Policy

Reviewer #2: No

Reviewer #3: No

---

## [Editor Report · Acceptance letter]

PONE-D-25-44441R1

PLOS One

Dear Dr. Ramsby,

I'm pleased to inform you that your manuscript has been deemed suitable for publication in PLOS One. Congratulations! Your manuscript is now being handed over to our production team.

Kind regards,

on behalf of

Dr. Satheesh Sathianeson

Academic Editor

PLOS One